High bycatch rates of manta and devil rays in the “small-scale” artisanal fisheries of Sri Lanka

Fernando Daniel daniel@blueresources.org 1 2 3
Stewart Joshua D. 2
1 Blue Resources Trust , Colombo , Sri Lanka
2 The Manta Trust , Dorchester , United Kingdom
3 Department of Biology and Environmental Science, Linnaeus University , Kalmar , Sweden
Jeffery Nicholas
Electronic publication date: 2021 Sep 8
Publication date: 2021
Volume: 9
Electronic Location ID: e11994
Received 2021 May 21; Accepted 2021 Jul 27
Copyright: ©2021 Fernando and Stewart
Copyright year: 2021
Copyright holder: Fernando and Stewart
License: This is an open access article distributed under the terms of the Creative Commons Attribution License, which permits unrestricted use, distribution, reproduction and adaptation in any medium and for any purpose provided that it is properly attributed. For attribution, the original author(s), title, publication source (PeerJ) and either DOI or URL of the article must be cited.
License URL: https://creativecommons.org/licenses/by/4.0/

Keywords: Mobula, Small-scale fisheries, Bycatch, Age structure, Size at maturity

Funding: Linnaeus University The Save Our Seas Foundation The Manta Trust The Shark Conservation Fund The Marine Conservation and Action Fund The Ocean Park Conservation Foundation Hong Kong [FH02_1920] The Tokyo Cement Group This work was supported by Linnaeus University, the Save Our Seas Foundation, the Manta Trust, the Shark Conservation Fund, the Marine Conservation and Action Fund, the Ocean Park Conservation Foundation, Hong Kong [FH02_1920], and the Tokyo Cement Group. The funders had no role in study design, data collection and analysis, decision to publish, or preparation of the manuscript.

==============================
Background

Expanding fisheries in developing nations like Sri Lanka have a significant impact on threatened marine species such as elasmobranchs. Manta and devil (mobulid) rays have some of the most conservative life history strategies of any elasmobranch, and even low to moderate levels of bycatch from gillnet fisheries may lead to significant population declines. A lack of information on life history, demographics, population trends, and fisheries impacts hinders effective management measures for these species.

Method

We report on mobulid fishery landings over nine years between 2011 and 2020 across 38 landing sites in Sri Lanka. We collected data on catch numbers, body sizes, sex, and maturity status for five mobulid species. We used a Bayesian state-space model to estimate monthly country-wide catch rates and total annual landings of mobulid rays. We used catch curve analyses to estimate total mortality for Mobula mobular, and evaluated trends in recorded body sizes across the study period for M. mobular, M. birostris, M. tarapacana and M. thurstoni.

Results

We find that catch rates have declined an order of magnitude for all species across the study period, and that total annual captures of mobulid rays by the Sri Lankan artisanal fishing fleet exceed the estimated annual captures of mobulids in all global, industrial purse seine fisheries combined. Catch curve analyses suggest that M. mobular is being fished at rates far above the species’ intrinsic population growth rate, and the average sizes of all mobulids in the fishery except for M. birostris are declining. Collectively, these findings suggest overfishing of mobulid ray populations in the northern Indian Ocean by Sri Lankan artisanal fisheries. We recommend strengthening the management of these species through improved implementation of CITES, CMS, and regional fisheries management actions. In addition, we report on the demographic characteristics of mobulids landed in Sri Lanka and provide the first record of M. eregoodoo in the country.

Introduction

Small-scale and artisanal fisheries are increasingly recognized as having an equal or greater impact on marine resources than large-scale industrial fisheries, with recent studies suggesting that small-scale fisheries in developing countries employ over 90% of all fishers and are responsible for half of all fisheries landings (Shester & Micheli, 2011; Kelleher et al., 2012; FAO, 2015; Crona et al., 2015). Yet these fisheries are frequently unmonitored, often unregulated, and represent a widespread challenge to enforcement of fisheries management regulations. The most commonly utilised gear type in small-scale fisheries is the gillnet; due to ease of manufacture, deployment, recovery, repair, its relatively low cost, and the wide range of species that can be effectively captured (Anderson et al., 2020; Berninsone et al., 2020). However, their versatility also makes gillnets responsible for an extremely large proportion of the world’s incidental capture and bycatch (Lewison et al., 2004; Peckham et al., 2007; Žydelis et al., 2009; Reeves, McClellan & Werner, 2011; Jabado, 2018; Arlidge et al., 2020). Further, because gill nets are often deployed for many hours or days at a time, the bycatch mortality associated with gillnet fisheries can be much higher than other gear types such as purse-seines and longlines (D’agrosa, Lennert-Cody & Vidal, 2000; Thorpe & Frierson, 2009; Broadhurst & Cullis, 2020). Efforts to phase out gillnets in favour of fishing gears with lower bycatch rates are often challenging (Squires, 2010; Rojas-Bracho & Reeves, 2013), particularly as alternatives generally result in lower capture and higher operational costs, a combination that is not favoured by low-income fishers or governments seeking to expand fishery revenue.

Sri Lanka, a small, developing island nation situated in the Indian Ocean, has a considerable proportion (27%) of its 21.4 million inhabitants living along its 1,770 km coastline (Emerton, Miththapala & Sutharkaran, 2021). With an Exclusive Economic Zone (EEZ) that covers an area greater than 517,000 km2, fish products are an important and dominant source of protein, comprising 50% of total animal protein consumed by the population (Ministry of Fisheries and Aquatic Resources Development, Sri Lanka, 2020). Fisheries are also an integral part of the culture and economy, directly employing over 224,610 marine fishers and contributing around 1.1% to GDP, resulting in focused efforts by the government to increase national fishing productivity (Government of Sri Lanka, 2019; Ministry of Fisheries and Aquatic Resources Development, Sri Lanka, 2020). These include the provision of fuel and boat subsidies, in addition to the establishment of new fishery harbours, fish processing and canning factories, and larger vessels to engage in deep sea trawling. Nonetheless, the fisheries industry is still heavily reliant upon small-scale and artisanal fisheries, a large proportion (56%) using gillnets or gillnets in combination with longline or ringnets to target yellowfin and skipjack tuna (Hewapathirana, Gunawardane & Ariyaratne, 2019).

While tuna and billfish species fetch the highest prices for fishers, the Sri Lankan gillnet fleet captures a wide variety of bycatch species including marine mammals, turtles, and elasmobranchs (Karunasinghe, Fernando & Samaraweera, 2000; Jayasinghe et al., 2019; Anderson et al., 2020). It is well documented that in small-scale fisheries, where almost all the catch is consumed or utilized, retained bycatch have a high value (Ardill, Itano & Gillett, 2011), and fisheries in Sri Lanka are no exception. While some bycatch is discarded at sea due to prohibitions on retention (e.g., marine mammals, turtles, whale sharks, oceanic white-tip sharks and thresher sharks), other species, most notably elasmobranchs, are retained and sold due to national and international demand for their meat and derivatives.

Manta and devil rays (genus Mobula) consist of 9 extant species found in tropical, subtropical and temperate waters worldwide (Notarbartolo di Sciara et al., 2017; Notarbartolo di Sciara et al., 2020; Hosegood et al., 2020). Collectively referred to as mobulids, they are among the largest of all rays and tend to favour pelagic habitats (Couturier et al., 2012). Similar to many other elasmobranchs, mobulid rays are highly susceptible to overexploitation due to their slow population growth rates driven by low fecundity, matrotrophic reproduction, large size at birth, slow growth rates, late maturity, and longevity (Couturier et al., 2012; Dulvy et al., 2014; Pardo et al., 2016). While significant progress has been made in describing the biology and ecology of mobulid rays over the last decade, information on the biology and life history of most mobulid species is still extremely scarce (Stewart et al., 2018a). The best-studied member of the genus is the reef manta ray (Mobula alfredi), mostly due to its tendency to aggregate in coastal environments. Studies on the oceanic manta ray (M. birostris) have also expanded in recent years, resulting in important baseline data on movements, feeding ecology, courtship, and mating (Burgess et al., 2016; Stevens, Hawkins & Roberts, 2018; Bessey et al., 2019; Beale et al., 2019; Trujillo-Córdova et al., 2020; Armstrong et al., 2020). However, the rest of the devil rays have received far less scientific attention, and consequently major knowledge gaps exist in reproductive biology, population connectivity, geographic range, movements, habitat use, and foraging behaviour (Couturier et al., 2012; Stewart et al., 2017; Stewart et al., 2018a).

International demand over the last decade for mobulid gill plates—the cartilaginous structures that filter plankton from the water column—has directly led to an increase in fishing effort and retention of these species in bycatch fisheries (Croll et al., 2016; O’Malley et al., 2017). This demand for dried mobulid gill plates, which are marketed as a form of Traditional Chinese Medicine in east Asia, was fuelled by a rapidly expanding middle class alongside the depletion of more desirable fish stocks (Graham-Rowe, 2011; Fabinyi, 2012; O’Malley et al., 2017). Mobulid rays are highly susceptible to incidental capture in a wide range of fisheries and gear types including gillnets, purse seines, trawl nets, and on occasion even long lines (Croll et al., 2016). Together with their conservative life history characteristics, this has led to a number of suggested population declines around the world (Ward-Paige, Davis & Worm, 2013; Dulvy et al., 2014; White et al., 2015; Lewis et al., 2015; Pardo et al., 2016), including in the western Indian Ocean (Pacoureau et al., 2021). Sri Lanka is considered one of the largest mobulid fisheries in the world (Ward-Paige, Davis & Worm, 2013; CMS Secretariat, 2015; Acebes & Tull, 2016; Marshall et al., 2019) due to the expansive fishing fleet that ventures far beyond the country’s EEZ.

To improve implementation of international agreements to protect mobulid rays (e.g., Convention on International Trade in Endangered Species of Wild Fauna and Flora (CITES) and the Convention on Migratory Species of Wild Animals (CMS)) and support further national action, we report on the catch and landing composition of mobulid rays encountered during surveys of domestic fish markets from 2010 to 2020. We use Bayesian state-space models to estimate the trends in catches across this period for each of four mobulid species and to estimate the total landings of mobulids in the surveyed markets. Additionally, we use demographic analyses of size and age distributions to infer total mortality and impacts of the fishery on mobulid rays. Finally, we report demographic data including sex composition and size at maturity, contributing to the limited available information on the reproductive biology of these species.

Methods

Study sites

We recorded mobulid ray landings over 1,346 surveys at 38 landing sites (combination of fishery harbours and beach landing sites) in Sri Lanka between May 2010 and February 2020 (Fig. 1). Data until March 2014 was collected by the project leader (DF), and from March 2014 to March 2020 by 15 research assistants with varying durations under the direct supervision of DF, who performed quality control checks on all data collected and entered by research assistants. One research assistant’s morphometric measurements of mobulid specimens were deemed to be unreliable based on several implausible values identified during quality control assessments and therefore all measurements collected or entered by that researcher were eliminated from length-based analyses. All species identification was confirmed by DF based on images collected in the field by research assistants.

Figure 1 Bathymetric (Amante & Eakins, 2009) map of Sri Lanka showing the continental shelf extent around the country, and locations of the 38 study sites.

Numbers in parentheses indicate multiple study sites in a given locality.

Sites encompassing both single-day (IDAY: Inboard Single-day Boats and OFRP: Outboard engine Fiberglass Reinforced Plastic Boats) and multi-day (IMUL: Inboard Multi-day Boats) vessels were monitored, providing us access to specimens caught both nearshore and offshore. However, given the proximity of the continental shelf drop-off to shore (Karunarathna et al., 2020), pelagic species are often encountered even by the single day vessels. Unless specimens were counted as they were offloaded from a specific vessel, it was not possible to assign catches to single or multi-day vessels. Consequently, we considered only combined landings for the purposes of this study and do not distinguish between catches from single and multi-day vessels.

Data collection

Whenever possible, unstructured and informal interviews were conducted with fishers, auctioneers, and middlemen at the study sites to gather anecdotal information on mobulid fisheries, vessels and fishing grounds (GPS coordinates), gear types, catch seasonality, utilisation, and trade. We excluded implausible reported GPS coordinates that were on land. Trade of gill plates as reported by Sri Lanka to the Convention on International Trade in Endangered Species of Wild Fauna and Flora (CITES) were downloaded from the online trade database (CITES Trade Database, 2021).

Mobulid landings, demographic analyses, and CPUE

We identified all recorded individual mobulid rays to species level using the Guide to the Manta and Devil Rays of the World (Stevens et al., 2019). When possible, we recorded the disc width (DW) and disc length (DL) of individuals to the nearest centimetre. Due to constraints on the size of the opening to ice holds onboard fishing vessels, most rays were landed cut in half (vertically), or in several pieces for the largest bodied species (e.g., large M. birostris or M. tarapacana). In such cases we derived measurements from each segment and added them together to reconstruct a full DW.

Sexual maturity for males were recorded in two categories: immature having undeveloped claspers not extending beyond the pelvic fin and mature with fully calcified claspers extending beyond the posterior margin of the pelvic fin. We were unable to determine the maturity of females since the internal reproductive organs were usually removed prior to storage in the ice holds to slow down the rate of decomposition. Some individual specimens (including males) could not be sexed as their entire reproductive organs were cut out during gutting. Other practical complications included days when large numbers of rays were landed and piled on top of each other making it impossible to accurately identify the gender of each individual specimen.

To account for potential data recording errors, we performed a regression of DW to DL and excluded from the dataset any entries where a recorded DW was more than three standard deviations from the mean expected DW value based on the recorded DL. This resulted in the exclusion of one M. birostris, ten M. mobular, and three M. tarapacana data points. We estimated the size at maturity for males of each species by fitting a logistic regression to the DW and maturity data and calculating the size at which a male has a 50% chance of being sexually mature (DW50). We performed logistic regressions in R (R Core Team, 2020) with the glm function.

Due to the social dynamics of the fish markets and the speed at which mobulids were unloaded from vessels, purchased, and moved out of the market, it was not possible to collect any standard metrics of effort such as number of boats or number of fishing days in relation to the corresponding number of landed mobulids. Furthermore, since mobulid rays are not as valuable as target species such as tuna, the capture dynamics of mobulids are extremely complex. This makes it challenging to conceptualize, let alone collect, a valid metric of effort that would explain landing rates and allow for a reliable catch per unit effort (CPUE) estimation. Instead, we used the total number of registered single-day and multi-day vessels per year reported by the national Ministry of Fisheries and Aquatic Resources Development, as a coarse proxy for overall fishing effort in Sri Lanka (Ministry of Fisheries and Aquatic Resources Development, Sri Lanka, 2020).

Catch time series analysis

Since the fish markets were sampled irregularly throughout the study period, we chose to model trends in landings using monthly estimates of daily mean landings. In other words, the model estimated a mean landing rate for each month, around which daily, observed landings were distributed. We modelled landings as count data from a negative binomial distribution and used a state-space model to account for both observation error and true variability in the underlying process (i.e., actual increases and decreases in catch rates that deviate from the overall trend). We selected the negative binomial distribution to explain catch data as initial models using the Poisson distribution resulted in overdispersion in the data relative to model estimates. The model was described by the equations: (1) lnC ˆs,t= lnC ˆs,t−1+μs,t−1

where C ˆ is the overall mean daily catch rate (landed mobulids per day) for month t and species s, which is determined by the mean daily catch rate for the previous month (C ˆs,t−1) multiplied by the change in catch rate μs,t (additive in log space). The species-specific monthly change in catch rate μs,t is defined by: (2) μs,t∼Nμ ˆs,σts

where μs,t is drawn from a normal distribution with mean μ ˆs where μ ˆ is the overall trend in catches for species s and σts is the process error term allowing for monthly variation around the overall trend, estimated separately for each species. Each monitored fish market had a hierarchically distributed persistent mean random effect throughout the time series in order to estimate catches at markets during unsampled periods, such that: (3) M ˆm,s∼N0,σms

where M ˆ is the mean market random effect for each market m and species s, and σms is the process error term for the market effects, estimated separately for each species. The monthly market effect was then hierarchically distributed around the mean market effect: (4) Mm,s,t∼NM ˆm,s,σms,m

where M is the market random effect for each market m and species s in month t, and σms,m is the process error term defining the variability in the monthly market random effect around the mean market random effect, estimated separately for each species and market. Put simply, this allowed a market to have, for example, generally higher catch rates than the country-wide mean (M ˆm,s) but flexibility for the monthly catch rates to vary around that general effect (Mm,s,t). The mean catch rate at each market was then calculated as: (5) lnCm,s,t= lnC ˆs,t+Mm,s,t

where C is the mean catch rate for market m and species s in month t. Note that the Market effect is therefore estimated in log space. Finally, the observed catches are distributed around the mean catch rates using the negative binomial distribution such that: (6) catchm,s,t∼NegBinpm,s,t,nm,s,t

(7) nm,s,t=Cm,s,t∗pm,s,t1−pm,s,t

where catch is the observed catch at market m and species s in month t, p is the model-estimated probability parameter for the negative binomial distribution for market m and species s in month t, and n is the dispersion parameter for the negative binomial distribution, calculated as a function of the mean catch rate C and probability parameter p.

We did not include market random effects for M. thurstoni, as only 93 specimens were recorded over the full 10-year time series and were only recorded at 10 of 38 fishing ports. Instead, we estimated a single country-wide mean monthly catch rate and applied that to all markets for M. thurstoni. We did not include M. kuhlii in the catch estimation models as only 59 specimens were recorded in total. We conducted posterior predictive checks to assess the fit of our final model, in which we took draws from a random subset of the model estimated means for each month, species, and market and compared these to observed catches. We ran the model with JAGS (Plummer, 2003) using 3 chains with 400,000 iterations each, a burn-in of 200,000, and thinning of 240. We assessed convergence of the model using visual inspection of chain convergence and autocorrelation plots, and Gelman–Rubin diagnostics (Gelman & Rubin, 1992).

In order to estimate the total annual catch of each mobulid species we took random draws from the model-estimated mean catch for each month, species, and market using the estimated negative binomial parameters (similar to the posterior predictive checks). We sampled across the full posterior distribution for each month, species and market and drew n samples for each month, where n was the estimated number of fishing days that take place in that calendar month. To calculate fishing days, we excluded public and religious holidays from the number of calendar days that occur in each month and averaged this value across markets, as markets in different regions of the country often observe different religious holidays.

We then summed the posterior draws for the entire month and across all markets for each species to generate a monthly total for each species across all markets. Doing this for each posterior draw allowed us to generate a distribution of monthly total catch rates for each species, incorporating the uncertainty from the model fits into our estimates of total catch. Finally, we summed catches across all months within complete years covered by the study period (2011–2019) to generate annual estimates of total catch.

In addition to estimating total annual catch of each species using the above model, for comparison we conducted a simple extrapolation: (8) x ¯s,y=xs,yny∗Fy∗m

where x ¯ is the estimated total catch of species s in year y, x is the recorded number of landed mobulid rays of species s in year y, n is the number of survey days in year y, F is the estimated number of total fishing days in year y, and m is the total number of surveyed landing sites throughout the study period.

Size trend analyses

We also assessed annual changes in the average size of each mobulid species landed, as changes in size and age structure can be an indicator of fishing impacts on a population (Sharpe & Hendry, 2009; Enberg et al., 2012). We aggregated DW’s within years for each species and estimated the linear trend in mean DW using a simple Bayesian linear regression coded and run in R using JAGS (Plummer, 2003). A significant portion of the width data were collected by research assistants, and we assume that measurements collected by assistants are likely to have more observation errors than measurements collected by the project leaders. To account for this, we included an additive error term to our linear models for DW data collected by the research assistants. This additional error term was estimated by the model and allowed the data collected by research assistants to be normally distributed around the mean annual DW with a standard deviation that was greater than or equal to the standard deviation of width data collected by the project leader. In short, this explicitly introduces into the model the expectation that the research assistant’s data were less precise than project leader data but allows for the situation where both data sets were equally precise. It does not, however, allow for the research assistant data to be more precise than project leader data in estimating annual mean DWs. For linear models, we ran 3 chains of 100,000 iterations each, a burn-in of 50,000, and thinning of 60.

Total mortality estimation for mobula mobular

A length-at-age relationship has previously been estimated for M. mobular (formerly Mobula japanica) using vertebral band pairs (Cuevas-Zimbrón et al., 2013). This relationship was subsequently used to convert DW data into age distributions, allowing for the estimation of total mortality in a fished population in Baja California, Mexico (Pardo et al., 2016). We replicated the methods applied in Pardo et al. (2016) to create a catch curve for M. mobular in the Sri Lankan fishery. First, we converted DW data to ages using the estimated parameters from the Von Bertalanffy growth curve with strong priors from Pardo et al. (2016). We chose to use growth parameters estimated using strong priors as opposed to weaker or uninformative priors because few large individuals were sampled in their study and the parameter values estimated using strong priors were more consistent with expectations of maximum size for the species. We used regression estimators to calculate total mortality Z in order to remain consistent with Pardo et al. (2016) and provide comparable estimates of mortality for the same species in a different fishery. Pardo et al. (2016) used an unweighted regression estimator of Z and truncated the right tail of the age distribution (although they also compared estimates of Z without right truncation). However, simulation studies of total mortality estimators have suggested that unweighted regression estimators can be biased and warn against using them for analysing catch-curve data (Smith et al., 2012). Instead, we used the recommended approach of weighted regression estimators with no right truncation and considered the age of full recruitment to be the age of maximum catch (Smith et al., 2012).

Following Pardo et al. (2016) we bootstrapped our age distributions to randomly include 80% of the data in each bootstrap run. We conducted 20,000 bootstrap runs, identified the peak age abundance within each subset of the data, and used the R package FSA (Ogle, 2017) to fit a weighted regression estimator to the natural log of the number of individuals in each age class beginning with the age of maximum catch. In addition to estimating Z for all years combined, we conducted this analysis separately for width data collected in 2011, 2012, 2013, 2017, 2018, 2019, and 2020 to generate annual estimates of Z and evaluate whether total mortality has changed over time.

Results

Dynamics of the fishery

Mobulid fisheries

Anecdotal information from fishers revealed that prior to 2010, mobulids were often released due to low value and demand in comparison to other species. During this study, even as late as 2012, it was observed that on rare occasions mobulid rays were being landed without their heads as fishers had discarded them at sea to reduce space taken up in their holds and assuming that traders were only interested in the meat. However, as fisher knowledge of the mobulid gill plate trade and demand grew, the rays were landed whole.

Nonetheless, as mobulid rays are largely secondary catch, whether they were retained depended on multiple factors, including the amount of target species captured, at which point during the trip the mobulid rays were captured in the nets, and how much freezer space was available. Due to limited capacity onboard vessels and the lack of refrigeration, fishers often stated that any mobulids captured when heading out to sea were generally discarded in favour of catching target species, and only retained them when returning back to shore in order to fill up their holds to maximise profits and cover expenditures. Many fishers also claimed that mobulids are “heaty”, melting the ice onboard faster than other fish.

Furthermore, if mobulid rays were spotted in the water, nets would not be laid out as the rays tend to entangle themselves due to their forward swimming nature coupled with the commonly observed summersault behaviour exhibited when they encounter obstacles, including gillnets, entrapping them even further. All this requires additional time for fishers to remove them from the nets and repair any damage. There is also the potential for the entire net to be lost as many mobulid species, except for M. birostris, tend to swim in schools resulting in multiple individuals getting captured at the same time, dragging down the nets under their accumulated weight. The fishers maintain that this is the reason they have to haul their nets in quickly when mobulids are caught, since once they die, their weight compounded with their negative buoyancy can result in fishers being unable to bring up the nets using only their hands, or sometimes even with a winch. This results in the nets being cut and abandoned, causing significant financial losses considering that the nets are usually approximately 50% of the investment placed into launching a fishing vessel in Sri Lanka. Several times during the surveys we observed large M. birostris specimens being towed in the nets back to port as it was too strenuous for the fishers to haul the individual onboard at sea. Fishers also stated that the extremely rough skin of M. birostris caused damage to their hands and were less appealing to bring onboard.

Vessels and fishing grounds

Fishing was predominantly conducted by three types of vessels; single-day boats fitted with either outboard motors (maximum 40hp) or small inboard diesel engines that fish in coastal waters for durations of up to 24 h, and multi-day vessels with diesel inboard engines that fish offshore and on occasion in the high seas for anywhere between around 14 to 90 days, depending on weather conditions and catch. However, even in the case of single day vessels, there was a high likelihood that they were fishing in deep waters due to the narrow continental shelf edge around Sri Lanka that drops to over 200 m in depth within 20 km of the shoreline, or in some areas such as Trincomalee on the east coast, dropping to over 600 m within the harbour itself due to the presence of a deep-sea canyon. Only a few of these fisheries would have been focusing effort at inshore coral reef habitats, and fishing at offshore coral reefs, such as Bar Reef on the north-west coast, likely also took place.

We collected 78 reported GPS coordinates of fishing grounds from interviewed fishers and excluded four of these as they were situated on land (three points in Sri Lanka, and one point appearing on the border between China and Russia). Fishers in multi-day vessels often sailed great distances to reach the most productive fishing grounds, with some claiming that Sri Lankan waters were already over-fished, resulting in the need to venture into the high seas or even neighbouring EEZ’s including India, the Maldives, and sometimes as far as Indonesia, Seychelles, and the Chagos Archipelago. This is corroborated by reported GPS coordinates of catch locations from skippers of vessels that unloaded mobulids between 2011 and 2015 (Fig. 2). Fishers at some landing sites on the west coast (predominantly in Beruwela) mentioned that the Chagos Archipelago was especially good for shark fishing. Between 2010 and 2018, some fishers reported that they had shifted their fishing behaviour in recent years to avoid venturing too close to east Africa due to concerns over piracy surrounding the Somalian region.

Figure 2 Distribution of reported fishing locations.

(A) Map showing the Exclusive Economic Zones (EEZs) with points (in red) displaying locations of mobulid fishing grounds, collected between 2011 and 2015. (B) Bathymetric (Amante & Eakins, 2009) map of Sri Lanka and neighbouring country EEZ’s showing locations (in red) of mobulid fishing grounds, collected between 2011 and 2015. Implausible reported GPS coordinates (i.e., locations on land) were excluded.

Gear types

Most vessels that we surveyed carried gillnets or a combination of both gillnets and longlines, and reported deploying them depending on weather conditions, season, and species availability. Mobulid rays were landed as non-discarded bycatch of pelagic, or sometimes bottom-set, gillnet fisheries with mesh sizes ranging from 11.43 cm (4.5 inches) to 15.24 cm (6 inches) that were primarily targeting skipjack tuna (Katsuwonus pelamis), yellowfin tuna (Thunnus albacares), and billfish. Only 3 records were logged of potential capture by fishing lines (rather than nets): one M. kuhlii specimen by a longline, and two M. mobular specimens, one by handline and the other was documented at the landing site with a hook in its mouth. Across all surveys there were ten records of mobulid rays being captured using hooks (similar to a harpoon but primarily used to drag large fish along the boat deck) when encountered at the surface: six M. birostris, one M. mobular, and three M. tarapacana; all at landing sites along the southern tip of Sri Lanka and prior to 2015.

Catch seasonality

There appeared to be ambiguity among fisher claims regarding the effects of the monsoons on mobulid catch. Some suggested there was a correlation while others said there was no influence on the catch but rather a shift in numbers landed due to decreases of target species catch during monsoonal periods. Either way, both agreed that the numbers landed generally increased during the onset of the south-west monsoon (May to September) for ports along the west coast and during the north-east monsoon (October to March) for ports on the east coast. We note that there was no clear seasonal signal in the estimated mean landings rates from our state-space model, but we did not include a spatial component, and country-wide averages may have masked alternating peaks in landings between the east and west coasts of the country.

Multiple fishers on both sides of the island mentioned that the catch and presence of mobulid rays were strongly correlated with the presence of krill, and that while the rays preferred deeper waters, they (particularly M. birostris) would follow the krill to the surface. This information was corroborated by whale-watching operators based at the Mirissa Fishery Harbour in the South, who said they occasionally spotted manta rays feeding on krill at the surface.

Mobulid utilisation and trade

Only two of the surveyed communities (Negombo and Chilaw) were observed selling fresh mobulid meat. At both sites, M. tarapacana was the most valued, followed closely by M. mobular. Both M. thurstoni and M. kuhlii were considered the same as M. mobular (and often not identified as separate species) and only fetched lower prices due to their smaller size. Mobula birostris was almost never consumed fresh as it was said to have a “sandy” or “grainy” texture. When not consumed fresh (and any excess), the meat was removed off the cartilaginous structure and cut into strips, sun-dried, and sold as dry fish for consumption or animal fodder.

The highest value part of mobulid rays were their gill plates which were exported (see Table S1). These cartilaginous structures are removed from the heads, cleaned by detaching gill filaments and excess tissue, and then sun-dried. Due to the technical skills required to keep the plates intact during removal (larger, intact gill plates fetch higher prices), together with the contacts required to establish and maintain the supply chain and to avoid market saturation, each landing site had at most around five middlemen that purchased gill plates. These middlemen were often the same as those involved in the shark fin trade and often, both commodities went via the same supply chain. These middlemen either purchased the heads of the mobulids or the entire specimen. They would then extract the gill plates and resell any meat to retailers. The mobulids were always sold by fishers via auctioneers, and on days where few individuals were landed, they were auctioned off specimen by specimen. On days with high landings they were auctioned in bulk, by weight. When individual specimens were auctioned, depending on the landing site, they were either sold as one entire piece or in halves (as they were most frequently landed), or the head (containing the gill plates) separate from the rest of the body. The sale prices varied significantly day to day and appeared to be dependent on the number of rays landed on a given day and in relation to the days prior to that, and taking into account the number of middlemen (purchasing power) present during auctioning itself.

Mobulid landings and CPUE

Species composition, size frequency, and size at maturity

Most of our survey effort was focused at five of the primary landing sites: Mirissa fisheries harbour (n = 172 survey days), Valaichchenai fisheries harbour (n = 170), Negombo (n = 135), Valaichchenai landing sites (n = 145), and Valaichchenai fish market (n = 124). Nine additional sites had more than 20 days of survey effort (range 22–86 survey days), and the remaining 24 sites had fewer than 20 days of survey effort (range 1–14 survey days). Mobulid rays were recorded at 21 out of 38 monitored landing sites. A total of 6,516 individual specimens, comprising five species, were recorded over the 1,346 surveys (see Table 1). These included pups of two M. mobular (one female: DW 74 cm; one male: unmeasured), three M. tarapacana (two females: DW 132 cm and 149 cm; one unknown gender: DW 117 cm), and four M. kuhlii (three females: DW 35 cm, 49 cm, 62 cm, and 63 cm; one male: DW 63 cm). While there were other small individuals of similar sizes of each species (Fig. 3), they could not be confirmed to be pups (i.e., unable to confirm whether they were extremely small free-swimming individuals, or fetuses that had been aborted in the nets or during handling). The smallest pregnant M. mobular had a DW of 200 cm (see Fig. 4) (with another three pregnant with: DW of 205 cm, and two with a DW of 209 cm), M. tarapacana had a DW of 304 cm (with another: DW of 326 cm), and M. kuhlii had a DW of 128 cm (with another three: DW of 130 cm, 132 cm, and 138 cm). We note that in June 2020—after the sampling period included in our analyses—one M. eregoodoo specimen was recorded, bringing the total number of Mobula species in Sri Lanka to six.

Table 1 Demographic characteristics of mobulid rays landed in Sri Lanka.

	Mobula birostris	Mobula mobular	Mobula tarapacana	Mobula thurstoni	Mobula kuhlii	
Specimens recorded						
Females	107	1,598	388	41	18	
Mature males	22	670	115	8	18	
Immature males	73	1,077	224	32	11	
Unknown males	1	3	0	0	0	
Unsexed	96	1,550	387	12	12	
Total a	299 (4.6%)	4,898 (75.2%)	1,114 (17.1%)	93 (1.4%)	59 (0.9%)	
Size range (including pups)						
Females	180–478 cm	60–252 cm	132–326 cm	72-168cm	35–138 cm	
Mature males	396–449 cm	184–242 cm	212–314 cm	138–160 cm	101–124 cm	
Immature males	136–375 cm	62–222 cm	98–244 cm	65–145 cm	63–104 cm	
Unsexed	168–308 cm	71–236 cm	115–300 cm	98.5-115.5 cm	75 cm	
Size at 50% maturity (Fig. 4)						
Males	385.53 cm	202.51 cm	239.99 cm	142.78 cm	102.87 cm	
Proportion of immature						
% of males immature	76% of 96 specimens	62% of 1,750 specimens	66% of 339 specimens	80% of 40 specimens	38% of 29 specimens	
% of males smaller than DW50	86% of 56 specimens	67% of 431 specimens	75% of 112 specimens	79% of 34 specimens	32% of 28 specimens	
% of females smaller than DW50b	96% of 53 specimens	83% of 406 specimens	87% of 117 specimens	97% of 31 specimens	56% of 18 specimens	
Notes.

a This excludes 53 specimens (0.8%) that were not identified to species level.

b DW50 for females from Rambahiniarison et al. (2018): M. birostris: 448.0 cm; M. mobular: 217.8 cm; M. tarapacana: 264.8 cm; M. thurstoni: 163.6 cm and the size of the one recorded pregnant female specimen of M. kuhlii from Notarbartolo di Sciara et al. (2017): 116.8 cm. These figures are expected to be higher as they are based on DW50 and not the DW of the smallest pregnant individual, however previous studies have shown delayed onset between maturity and pregnancy (Rambahiniarison et al., 2018).

CPUE proxy

The number of registered IMUL and IDAY fishing vessels show an overall increasing trend; from 4,523 in 2010 to 5,833 in 2019, while the number of OFRP vessels remained fairly stable after an initial increase in 2011 (see Table 2). Based on informal surveys and canvassing during the study period, the majority of mobulid landings appeared to be associated with IMUL and IDAY vessels that fish in the high seas and throughout the EEZ, while some of the more coastal species such as M. kuhlii were more frequently landed by the OFRP vessels that tend to fish a few kilometres offshore.

Figure 3 Mobulid size distributions.

Frequency histograms represent the distributions of males, females, and unsexed individual mobulid rays landed across the entire study period.

Figure 4 Size at maturity of the five mobulid species landed in Sri Lanka.

Points indicate individual male rays classified as mature (1) or immature (0) based on clasper length and calcification. Coloured curves indicate the logistic regression fit for size at maturity. Vertical black lines indicate the estimated disc width at 50% maturity (DW50) for the five species (note the varying x-axis scale).

Table 2 The number of registered fishing vessels from 2010 to 2019 as reported by the Ministry of Fisheries and Aquatic Resources Development (MFAR, 2020).

IMUL are inboard [engine] multi-day boats operating within and beyond the EEZ, while IDAY are inboard [engine] single-day boats operating within the EEZ, and OFRP are outboard engine fiberglass reinforced plastic boats operating in coastal waters.

	2010	2011	2012	2013	2014	2015	2016	2017	2018	2019	
IMUL:	3,346	3,872	4,080	4,111	4,447	4,218	3,996	4,196	4,581	4,885	
IDAY:	1,177	1,120	890	802	876	719	786	868	918	948	
SUB-TOTAL:	4,523	4,992	4,970	4,913	5,323	4,937	4,782	5,064	5,299	5,833	
OFRP:	18,770	22,890	23,160	23,134	23,982	24,028	24,282	22,394	24,132	23,404	
TOTAL:	23,293	27,882	28,130	28,047	29,305	28,965	29,064	27,458	29,631	29,237	

Catch time series

State space models

The median value of the overall trend in catches across all sampled years was −0.012 (95% CIs [−0.048–0.021]) for M. birostris with an 81.7% probability that the mean trend in catches was negative; −0.013 (95% CIs [−0.038–0.015]) for M. mobular with an 86.8% probability that the mean trend in catches was negative; −0.010 (95% CIs [−0.031–0.018]) for M. tarapacana with an 83.6% probability that the mean trend in catches was negative; and −0.015 (95% CIs [−0.038–0.008]) for M. thurstoni with a 92.8% probability that the mean trend in catches was negative. In all of our posterior predictive checks, observed numbers of mobulids landed at a specific market fell within the prediction intervals based on model-estimated parameters (Figs. 5 and 6, and Figs. S1–S4).

Figure 5 Overall country-wide model estimated mean monthly catch rates by species.

These values represent the mean catch rates before market-level random effects are applied. The black line indicates the median estimate of the posterior distribution, and the coloured polygons represent the posterior credibility intervals ranging from the 50% Cis (darkest) to 90% Cis (lightest). Squares along the bottom indicate months with sampling effort (at any market). Open squares indicate zero landings of the respective species, and filled squares indicate one or more recorded individuals landed.

Figure 6 Catch trends for M. mobular from the landing sites with the six highest mean market random effects.

The black line indicates the median estimate of the posterior distribution, and the coloured polygons represent the posterior credibility intervals ranging from the 50% Cis (darkest) to 90% Cis (lightest). Squares along the bottom indicate months with sampling effort at each market. Open squares indicate zero landings of the respective species, and filled squares indicate one or more recorded individuals landed.

Our catch reconstructions based on predictions from model-estimated catch rates are reported in Table 3. Given the uncertainty around total catch estimates that use parameter estimates from unobserved periods in the model, we report both 50% and 90% credible intervals (rather than 95%) to avoid reporting extraordinarily high catch estimates at the far-right tail of the predicted distribution. Catch reconstructions from the model-estimated parameters and simple extrapolations from average mobulid landings generally agreed well. Total estimated countrywide catches of M. birostris (median estimates) ranged from a minimum of 1,025 (2017) individuals to a maximum of 7,961 (2014), although both 50% and 90% credible intervals suggest that total catches could be considerably higher. Median estimates of total M. mobular catches range from 11,258 (2017) to 98,059 (2011); M. tarapacana from 4,867 (2019) to 13,966 (2013); and M. thurstoni from 533 (2017) to 2,915 (2011). We did not reconstruct total catches from 2010 or 2020 as only a small portion of each year was sampled. The model estimated 2016 total catches although no simple catch extrapolations were made as no samples were collected during 2016. In the case of M. mobular, M. tarapacana and M. thurstoni, total estimated catches declined by an order of magnitude over the 10-year study period. Mobula birostris total estimated catches decline by approximately 75% from 2011 to 2018, but roughly doubled from 2018 to 2019 (Table 3).

Table 3 Annual catch reconstructions of mobulid rays in Sri Lanka.

Simple refers to the simple extrapolation described in equation 8. All other columns refer to the state-space model of fisheries landings: minimum and median values of the posterior distribution, and 50% and 90% credible intervals for each year and each species. Values represent the total model-estimated number of individuals landed across all sampled markets and landing sites. No data were collected in 2016, so the sample extrapolation was not possible in that year.

Species/Year	Simple	Minimum	Median	50% Cis	90% Cis	
Mobula birostris	
2011	5,714	610	4,461	2,736–8312	1,884–17987	
2012	3,705	548	3,691	2,214–7,086	1,506–13,112	
2013	5,217	532	4,317	2,650–8,049	1,854–16,426	
2014	5,537	1,447	7,961	4,975–14,296	3,482–29,459	
2015	0	104	2,691	1,467–5,398	877–11,336	
2016	–	62	1,831	929–3,905	496–8,863	
2017	502	62	1,025	611–1,925	398–4,067	
2018	829	285	1,217	803–2,121	603–4,350	
2019	2,885	713	2,785	1,920-4,810	1,421–10,018	
Mobula mobular	
2011	251,547	25,222	98,059	68,577–163,796	52,453–319,592	
2012	177,099	25,541	79,794	56,550–128,197	44,338–254,314	
2013	205,820	19,717	59,238	42,477–95,140	33,256–180,457	
2014	48,319	14,235	39,520	28,640-63,670	22,982–123,535	
2015	14,227	4,134	18,076	12,249–30,149	9,122–55,762	
2016	–	1,836	14,584	9,546–25,119	6,677–50,398	
2017	12,258	2,550	11,258	7,922–17,804	6,249–31,888	
2018	11,729	4,163	11,284	8,490–17,638	6,970–32,053	
2019	15,294	4,462	12,075	8,968–18,863	7,464–35,053	
Mobula tarapacana	
2011	32,568	3,611	13,023	8,852–21,861	6,743–47,131	
2012	30,875	4,059	13,729	9,335–23,313	7,179–42,619	
2013	32,723	3,321	13,966	9,332–23,491	7,327–46,587	
2014	18,064	3,872	12,866	8,961–22,434	7,098–43,663	
2015	9,485	1,313	8,192	5,515–14,626	4,069–30,786	
2016	–	990	7,304	4,865-12,802	3,491-26,391	
2017	4,622	1,519	5,966	4,165–9,923	3,130–20,078	
2018	3,697	1,699	5,293	3,635–8,823	2,883–17,926	
2019	4,347	1,148	4,867	3,510–8,405	2,740–17,768	
Mobula thurstoni	
2011	2,857	981	2,915	2,434–3,495	2,061–4,100	
2012	4,940	658	2,283	1,901–2,774	1,589–3,343	
2013	1,897	361	1,603	1,345–1,930	1,117–2,338	
2014	671	314	1,059	880–1,243	744–1,430	
2015	0	65	825	631–1,019	456–1,216	
2016	–	32	665	489–848	344–1,047	
2017	201	80	533	400–672	302–795	
2018	351	175	495	411–584	343–679	
2019	593	164	559	459–674	388–778	

Size trends

Bayesian linear models indicated a negative trend in the DW of landed individuals for all sampled species except for M. birostris (see Fig. 7). Mobula mobular (N = 956) demonstrated a decrease in mean DW of 2.8 cm per year (95% CI [2.1–3.4] cm), M. tarapacana (N = 285) a decrease of 2.4 (1.1–3.6) cm per year, and M. thurstoni (N = 68) a decrease of 2.2 (0.36–4.1) cm per year. There was a 100% probability of a decline in body size of M. mobular and M. tarapacana, and a 98.9% probability of decline in body size of M. thurstoni (i.e., 98.9% of posterior estimates of the slope were negative). Mobula birostris (N = 121), in contrast, exhibited no change in mean DW (median slope 0.5 cm increase, 95% CI: 3.5 cm decrease–4.6 cm increase; 40% probability of decreasing body size). The additional uncertainty in DW measurements estimated by the model for research assistants were generally negligible, although somewhat higher for M. birostris. The median additional error term for M. mobular was 0.5 cm (0.024–2.5 cm), for M. tarapacana one cm (0.036–4.4 cm), for M. thurstoni 1.65 cm (0.056–8.2 cm), and for M. birostris 10.7 cm (0.68–31.7 cm).

Figure 7 Size trends of mobulid rays captured in Sri Lankan artisanal fisheries.

For each panel, the open circles represent the recorded disc widths of individual (A) Mobula birostris (N = 121), (B) M. mobular (N = 956), (C) M. tarapacana (N = 285), and (D) M. thurstoni (N = 68). Observations are binned by year and jittered on the x-axis for clarity. The black line in each panel represents the regression line generated from the median model estimated parameters. The coloured lines represent the full Bayesian posterior distributions of estimated regression parameters, with each line representing one paired combination of slope and intercept from the posterior distribution.

Total mortality estimates for mobula mobular

The size-to-age conversion using the Von Bertalanffy growth equation produced an age range similar to M. mobular in Pardo et al., 2016 (maximum age 11.73 years). A small proportion (0.049, n = 40) of M. mobular in this study were below age zero, which is consistent with occasional observations of foetuses aborted by pregnant females during capture. Our catch curve analysis including all individuals across all years produced a median estimate of total mortality Z = 0.671, with 95% bootstrapped confidence intervals of 0.359 –0.909. Median estimates of Z for individual years ranged from 0.256 (2017) to 0.795 (2019) (Fig. 8, Table 4). The majority of bootstrap estimates fell above the 0.108 upper confidence interval for maximum intrinsic rate of population growth estimated by Pardo et al. (2016) for M. mobular. Estimates from 2013 of Z had the highest percentage (10.1%) of bootstrap estimates fall below 0.108, followed by 2012 (3.7%). All other years had 0–0.1% of bootstrap estimates below 0.108, which suggests that the regional M. mobular population is experiencing mortality rates that far exceed the species’ intrinsic population growth rates.

Figure 8 Total mortality estimates for Mobula mobular.

Shaded violin plots indicate the bootstrapped distributions of total mortality (Z) estimates from catch curve analyses using annual disc width records and disc widths from all years combined (All). Inset boxplots indicate the median (horizontal black line) 50% (white box) and 95% (vertical black line) bootstrap intervals. The solid horizontal line near the bottom indicates the median estimate of rmax for M. mobular from Pardo et al. (2016), and dashed horizontal lines indicate the 95% confidence intervals of rmax. Numbers along the top indicate the sample sizes used to generate each estimate and bootstrap distribution.

Table 4 Estimates of total mortality (Z) for Mobula mobular.

Bootstrap median estimates and 95% intervals for Z calculated using disc widths across all surveys (All) or using disc widths from individual years. P(Z > 0.108) indicates the proportion of bootstrap estimates that were greater than the upper 95% confidence interval for rmax (0.108) estimated by Pardo et al. (2016). We interpret that as the probability that total mortality exceeded the species’ maximum intrinsic rate of population growth in a given year.

Year	Median	95% Interval	P(Z > 0.108)	
All	0.671	0.359–0.909	1	
2011	0.763	0.594–1.095	1	
2012	0.319	0.091–1.792	0.963	
2013	0.481	0–0.693	0.894	
2017	0.256	0.134–0.578	0.999	
2018	0.561	0.492–0.647	1	
2019	0.795	0.519–0.934	1	
2020	0.548	0.447–0.692	1	

Discussion

This study represents the first comprehensive evaluation of the mobulid fishery of Sri Lanka and one of the most extensive and detailed examinations of mobulid bycatch in any fishery worldwide. It provides an understanding of the scale of bycatch from this fishery and its probable impact on mobulid populations across the Indian Ocean. We found multiple lines of evidence suggesting that mobulid populations accessed by Sri Lankan fisheries are being exploited at unsustainable levels, including an extremely high annual catch of mobulids, decreasing overall catch rates in four species, high proportions of immature and juvenile individuals across four species, decreases in body sizes in landed individuals of three species, and a catch curve analysis of M. mobular indicating a total mortality rate far above the species’ intrinsic maximum population growth rate. Taken together, these findings suggest ongoing overfishing of mobulid populations in the northern Indian Ocean and, most likely, associated population declines (Pacoureau et al., 2021). This emphasises the urgent need for improved implementation of global conservation measures, along with better regional and domestic fisheries management aimed specifically at protecting mobulids, and other similarly vulnerable species.

Catch trends

Our catch reconstructions from state-space model fits reveal that Sri Lanka is one of the largest mobulid landing nations in the world with median estimates of total catch landings between 2011 and 2019 indicating more than 1,000 M. birostris, 11,000 M. mobular, 5,000 M. tarapacana, and 500 M. thurstoni landed annually at the 38 fishing harbours, landing sites, and markets sampled in this study. Our catch reconstructions suggest that annual mobulid landings in Sri Lanka have declined by an order of magnitude over the past decade, and that total catch of mobulids in the early 2010s may have exceeded 100,000 animals per year. There is cause to interpret these early estimates of total catch with caution. First, the majority of the 38 landing sites in this study have only been sampled since 2017 (Fig. S5). Our state-space model of catch rates estimates a country-wide mean monthly catch rate and overall trend in catch rates, along with a persistent random effect (or offset from the mean) for each sampled market. Therefore, catch rates are estimated for all markets for the full survey period, and catch rates for the majority of markets are estimated without data for 2010–2016. This assumes that all markets followed the overall country-wide trend, which is informed primarily by catch rates at 8 markets that were sampled relatively consistently over the full study period; an assumption that may or may not be valid. Second, several of the markets sampled during the full study period were known to have some of the highest overall mobulid catch rates (e.g., Negombo and Mirissa), which is why they were initially targeted for sampling. Declines in catch rates at these markets may therefore bias the overall catch reconstruction, inflating early estimates of total catch. Nevertheless, the markets with continuous sampling throughout the study period demonstrate major declines in catch rates (e.g., Figs. 5–6), and even the minimum catch estimates from the total catch reconstruction demonstrate five- to ten-fold declines for most species. We also note that, despite the very high upper credible intervals of the state-space model catch reconstructions, the model estimates and simple extrapolations are generally closely aligned, and the state-space model posterior predictive checks correspond closely to observed catches. These 38 sites include 10 of the 21 national fishery harbours where both single- and multi-day vessels land their catch in Sri Lanka, and therefore these total catch reconstructions likely approach the correct order of magnitude of total mobulid landings throughout the country. However, given there are a further 883 documented smaller fish landing sites (Ministry of Fisheries and Aquatic Resources Development, Sri Lanka, 2020) in Sri Lanka, it may be realistic to consider the upper bounds of the credible intervals as plausible total catch estimates for the country.

The magnitude of these total catch estimates—even the lower estimates from 2017 onwards—are exceptionally high considering the assumed small population sizes for mobulids (Stewart et al., 2018a). To put these catch numbers in context, it was estimated that Indonesia landed between 1,050 and 2,400 M. birostris in their target fishery in 2002 (Dewar, 2002; Lewis et al., 2015), Peru landed 1,985 mobulid rays over one year at their largest mobulid landing site while estimates of the northern Peru region suggest ∼8,000 individuals per year (Croll et al., 2016; Alfaro-Cordova et al., 2017). The estimated global tuna purse seine mobulid bycatch is approximately 13,085 individuals per year (Croll et al., 2016). Therefore, despite Sri Lanka’s fishery being classified as primarily small-scale artisanal, we would argue that it has significant impacts on mobulid rays in the northern Indian Ocean that exceed the scale of industrial fishery impacts in the region. Furthermore, the decline in catch rates despite an increase in the number of registered offshore fishing vessels and a stable trend in the coastal vessels suggests that there may have been an even larger decline in catch per unit effort during the ten-year study period, although we acknowledge that the number of registered vessels is an imperfect proxy for fishing effort. Inferences about the status of fished populations made using catch-only data are often negatively biased and less accurate than stock-assessment methods that incorporate fishing effort, indices of abundance, age or length structure, and other biological data (Carruthers, Walters & McAllister, 2012). As we note here, effort data for this fishery is complex and inadequate for inclusion in more integrated stock-assessment approaches, and biological data on mobulid rays, including length-at-age relationships for all species but M. mobular, is extremely sparse. Improving and collecting appropriate effort data in complex artisanal fisheries contexts would allow for a more accurate accounting of catch per unit effort for these species and may facilitate more robust stock-assessment approaches in the future. In addition, genomic tools such as close-kin mark recapture (Bravington, Skaug & Anderson, 2016) could provide estimates of abundance to further contextualize these catch estimates.

Three additional pieces of evidence indicate overfishing of the mobulid populations that are accessed by Sri Lankan fishers. First, our catch curve analysis of M. mobular indicated total mortality rates that far exceed estimates of the species’ intrinsic maximum population growth rates (rmax). These estimates fluctuated across years, although much of this variability may be due to differences in sample size between years. Regardless of whether the catch curve analysis was conducted at annual intervals or with combined age data across years, mortality rates ranged from roughly two to six times higher than the upper bound of rmax estimates for the species (Pardo et al., 2016). This suggests that fisheries mortality for the regional population of M. mobular is exceptionally high and that the population is being overfished, which is consistent with both declining catch rates and the magnitude of the total catch relative to expectations about population size for the species. Second, we found strong evidence for a decline in average body size in three mobulid species. Fished populations of elasmobranchs often exhibit decreases in size structure, even in cases where fishing gear is not selective for larger size classes (Stevens, 2000). Because size-at-age relationships are only available for M. mobular, we were not able to conduct catch curve analyses for any other species. However, the steady decline of average disc width in M. mobular, M. tarapacana, and M. thurstoni of approximately 1–2% per year is similar to size and weight declines of other heavily exploited elasmobranchs (Baum & Myers, 2004; Clarke et al., 2013). While change in average size is a very coarse metric of size structure, these changes seem to be driven by a decreasing number of larger individuals in our samples (Fig. 7), which would be consistent with high mortality leading to a truncated age distribution. In contrast, we found no change in the mean disc width of M. birostris across the study period. Given that the vast majority of M. birostris landed in Sri Lanka are juveniles (76% of males and up to 96% of the females), we would not necessarily expect a decline in average body size even if the population is experiencing high levels of mortality, as the average size of M. birostris landed in Sri Lanka (278.6 cm) is not far from the lower bound of the estimated size at birth (∼180–200 cm) (Rambahiniarison et al., 2018; Stevens et al., 2019). The frequent landings of juvenile M. birostris are concerning given the paucity of information on their juvenile life stage, early life history, and nursery habitats, and suggest that Sri Lankan fishers may be accessing a nursery habitat for the species. Mobula birostris are rarely encountered in shallow, coastal waters around Sri Lanka, which may indicate that juveniles use an offshore or pelagic nursery habitat, similar to one described in the northern Gulf of Mexico (Stewart et al., 2018b).

Declining population trends were further corroborated by gill plate middlemen who reported that fewer individuals are being landed in recent years, and also by fishers who suggested that mobulid (and shark) bycatch in gillnets have declined over the past two decades. The rapid growth in the gill plate trade driving demand for mobulids, coupled with declines of more desirable fish species including tuna, billfish, and sharks, further suggests that fishing effort for mobulids was likely increasing during the study period and that the decline in CPUE may have been even higher than the decline in total catches estimated by our models. Beyond recorded landings, discards at sea may also be a significant source of unrecorded and unreported mortality of mobulids in Sri Lankan fisheries. Fishers reported discarding mobulids when heading out to sea in favour of more valuable target fish. While there is currently no information on the post release survival probability of mobulids in gill nets, it is likely lower than survival rates in purse seine and logline fisheries (Mas, Forselledo & Domingo, 2015; Francis, Jones & Library, 2016; Ellis, McCully Phillips & Poisson, 2017). Evaluating post release survival rates of mobulids in the Sri Lankan gill net fishery and other small-scale fisheries employing gill nets will be important in understanding the full impacts of these fisheries on mobulid populations.

Demographics and ecological inferences

The size ranges of mobulids reported here were largely consistent with fishery-based demographic characterizations of mobulids elsewhere in the Indo-Pacific (Notarbartolo-di Sciara, 1988; White et al., 2006; Rambahiniarison et al., 2018; Stevens et al., 2019). The exception was M. kuhlii, for which we report a new maximum disc width of 138 cm (previously 122 cm: Stevens et al., 2019). The DW at which 50% of males were sexually mature (DW50) reported here was within 10–15 cm of previous studies in the Philippines and Indonesia for all species (see Table 5). We also report the first DW50 for M. kuhlii, which was approximately 10 cm smaller than the suspected size at maturity from the eight individuals recorded in Indonesia (White et al., 2006). The sex ratios were approximately 1:1 for all species except for M. kuhlii, which had a roughly 1.5:1 male:female ratio, although this could be due to small sample sizes (Fig. 3). Interestingly, while landings of most species occurred throughout Sri Lanka, M. kuhlii was landed most often at ports in the northwest of the island and along the west coast. Mobula kuhlii were also predominantly captured by nearshore, coastal vessels, and a much higher proportion of M. kuhlii were mature when compared to other species. This may indicate a more restricted, coastal distribution than the other mobulid species throughout the shallow straits between Sri Lanka and India.

Table 5 The size at maturity at which 50% (DW50) of males are mature in this study (Fig. 4) compared with the other two published datasets from the region.

	DW 50
(from this study)	DW50Philippines
(Rambahiniarison et al., 2018)	DW50 Indonesia
(White et al., 2006)	
Mobula birostris	385.53	381.9	375.2	
Mobula mobular	202.51	205.8	201.6	
Mobula tarapacana	239.99	252.1	248.6	
Mobula thurstoni	142.78	158.4	153.8	
Mobula kuhlii	102.87	–	115-119a	
Notes.

a Range at which maturity was attained (n = 8 samples; insufficient to determine DW50).

Several interviewed fishers described mobulid rays conducting vertical migrations in search of krill. Diel vertical migrations have been recorded in M. birostris from Peru and Mexico (Stewart et al., 2016; Andrzejaczek et al., 2021), and euphausiids are known to be an important diet component for mobulids (Notarbartolo-di Sciara, 1988; Couturier et al., 2012; Rohner et al., 2017; Stewart et al., 2017). Other large planktivores, such as blue whales, are frequently encountered at the edge of the continental shelf feeding on sergestid shrimp (a predator of euphausiids) and euphausiids that occur at depths of >300 m (De Vos et al., 2018). Given that fishers predominantly deploy their nets at night, it is plausible that the mobulid rays are following the diel migrations of euphausiids and encountering surface set gill nets, similar to the capture patterns reported in the Philippines (Rohner et al., 2017).

There was ambiguity in responses of fishers regarding whether the seasonal monsoons in Sri Lanka affected mobulid catch, with some stating they did and others not. Acoustic tagging studies of reef manta rays in the Maldives, just south of Sri Lanka, have recorded east–west migrations of the population between monsoon seasons to take advantage of monsoonal upwelling that alternates from the eastern to western sides of the atolls (Harris et al., 2020). Similar monsoonal upwelling and productivity cycles occur on either coast of Sri Lanka (Silva & Davies, 1987; Vinayachandran, 2004; Suresh et al., 2016; Jinadasa et al., 2020), and it is possible that mobulids may be exhibiting similar resource-driven seasonal migrations. Future studies could evaluate the spatial distribution of catches across seasons in Sri Lanka, although we caution that the monsoons have significant impacts on fishing distributions due to strong winds and challenging weather conditions, which may obscure or confound ecologically-driven spatial distributions of mobulids.

Conservation and management of mobulid rays in Sri Lanka

While several studies have reported on proxies for mobulid population trends (e.g., changes in sighting rates (Ward-Paige, Davis & Worm, 2013; White et al., 2015; Pacoureau et al., 2021 and fisheries data Lewis et al., 2015; Pardo et al., 2016)), accurate data on abundance and absolute trends remain almost non-existent (Stewart et al., 2018a). Our study combines several of these approaches for a single location (catch rates, catch curve analyses, and size trends), but we remain unable to estimate absolute abundance trends. Nevertheless, it is clear that mobulid rays are unable to withstand sustained, high rates of fishing (Dulvy et al., 2014; Pardo et al., 2016) and Sri Lanka appears to be one more example where mobulid populations are likely in decline. However, since 2013, amid growing concern for their globally threatened status (see Table S2), some international precautionary measures were introduced.

The CITES convention provides the mechanism necessary to ensure that any trade of listed species across international borders is permitted if it will not be detrimental to the survival of the species. Given the 18-month delayed implementation to provide time to build capacity and introduce necessary regulations, these listings came into force for Manta spp. in September 2014, and for all Mobula spp. in April 2017. There was no indication from our surveys of fishers being aware of, or adhering to, CITES export restrictions. Illegal gill plate exportations from Sri Lanka, and interceptions of illegal shipments (mislabelled or lacking permits) in trade hubs such as Hong Kong, continue to be recorded (Hau, Ho & Shea, 2016; HK-SAR, 2020; Sri Lanka Customs, 2021).

Further, the inclusion of all mobulids on CMS requires Sri Lanka to enact domestic management policies, including prohibiting the take of mobulids at the national level, while also encouraging CMS Parties to improve regional coordination and cooperation to better protect these species. Nevertheless, the declines in landings that we report cannot be attributed to this convention given the poor implementation of CMS-listed elasmobranchs at a global level and the lack of any national legislative framework to implement CMS in Sri Lanka (Lawson & Fordham, 2018). Further, it should be noted that Sri Lanka as a Party to both Conventions would be unable to trade gill plates given that catch (“take”), and therefore trade, is prohibited under CMS Appendix I. However, the CITES Trade Database contains records of commercial gill plate trade, reported by Sri Lanka, between 2015 and 2020 (Table S1), likely in contravention of CMS obligations.

There is a clear need for immediate management action to curtail the high rates of fishing mortality of mobulid rays in Sri Lankan (and surrounding) waters. The national Department of Fisheries and Aquatic Resources (DFAR) have initiated a programme to limit gillnets (the primary gear type capturing mobulid rays). At present the prohibition is restricted to gillnets longer than 2.5 km operating by vessels in the high seas (beyond the EEZ), as required by the Indian Ocean Tuna Commission (IOTC) under Resolution 17/07 (IOTC, 2017). While the revised National Plan of Action for Sharks in Sri Lanka 2018-2022 mentions the establishment of a 2.5 km maximum length for gillnets for all multi-day (IMUL) and one-day (IDAY) vessels under its intended activities, this restriction has not yet been implemented (Ministry of Fisheries and Aquatic Resources Development, Sri Lanka, 2018) and may not have long-term consequences given the absence of a cap on the growing number of registered fishing vessels (see Table 2). Given the lack of any other national regulations to effectively manage these species, it is highly likely that overfishing of mobulid rays will continue in Sri Lankan fisheries in the foreseeable future unless management action is taken quickly.

The neighbouring countries of the Maldives and the Chagos Archipelago both prohibit fishing of mobulid rays and could provide some protection if mobulid population distributions extend across the northern Indian Ocean. However, with far-ranging Sri Lankan fisheries that access the High Seas and at times even illegally fish within the sovereign waters of neighbouring nations or MPAs (DFAR, 2013; BIOT, 2020; Kilgour, 2020), along with intensive and expanding fishing pressure from other Indian Ocean countries such as India (Parappurathu et al., 2020), it is unlikely that regional spatial protections implemented by other countries will provide sufficient refuge from fishing pressure to prevent population declines. The mobulid conservation and management measure (Resolution 19/03) introduced by the IOTC in 2019 (IOTC, 2019) provides a regional framework for reducing mobulid fisheries captures. The measure includes a retention ban for mobulid rays that will come into effect for all fisheries (including artisanal) by January 2022. At present it is enforced only for commercial fisheries, with delayed implementation for artisanal fisheries to enable time to increase awareness and improve mitigation strategies. Given insufficient enforcement of CITES and the lack of implementation of CMS in curbing mobulid landings in Sri Lanka, it remains to be seen whether the IOTC measure can better address these impacts.

Conclusions

Based on the apparent population declines of mobulid rays in the Indian Ocean (Pacoureau et al., 2021) and the extremely high rates of fishing mortality that we report here, it is clear that implementation and—perhaps more importantly—enforcement of fisheries management actions are urgently needed for mobulid rays in Sri Lanka and the northern Indian Ocean.

Supplemental Information

Supplemental Information 1 Raw data of mobulid ray landings in Sri Lanka

Mobulid raw data from 2010 to 2020 for Sri Lanka

Click here for additional data file.

Supplemental Information 2 Reported trade by Sri Lanka (CITES Trade (CITES Trade Database, 2021) of CITES Appendix II listed mobulid rays between 2010 and 2020

HK = Hong Kong; LK = Sri Lanka; T = commercial trade; W = Wild.

Click here for additional data file.

Supplemental Information 3 The global (IUCN Red List Assessment Results, 2021) and regional (Arabian Seas and Adjacent Waters - (Jabado et al., 2017)) IUCN Red List categories for mobulid rays

Click here for additional data file.

Supplemental Information 4 Posterior predictive checks for Mobula birostris landings

Histograms (in grey) represent the model-predicted distribution of possible catches at a given market or landing site and month. Vertical dashed lines indicate the observed landings in the respective month for each landing site. Dashed lines are jittered on the x axis for clarity.

Click here for additional data file.

Supplemental Information 5 Posterior predictive checks for Mobula mobular landings

Histograms (in grey) represent the model-predicted distribution of possible catches at a given market or landing site and month. Vertical dashed lines indicate the observed landings in the respective month for each landing site. Dashed lines are jittered on the x axis for clarity.

Click here for additional data file.

Supplemental Information 6 Posterior predictive checks for Mobula tarapacana landings

Histograms (in grey) represent the model-predicted distribution of possible catches at a given market or landing site and month. Vertical dashed lines indicate the observed landings in the respective month for each landing site. Dashed lines are jittered on the x axis for clarity.

Click here for additional data file.

Supplemental Information 7 Posterior predictive checks for Mobula thurstoni landings

Histograms (in grey) represent the model-predicted distribution of possible catches across all markets for a given month. Vertical dashed lines indicate the observed countrywide landings in the respective month. Note that the M. thurstoni component of the state-space model did not have market-level effects, as landings were rare compared to other mobulid species, and estimating market-level effects was therefore not feasible. Dashed lines are jittered on the x axis for clarity.

Click here for additional data file.

Supplemental Information 8 Catch rates and sampling of markets and landing sites

The ridges in a-c represent the posterior distributions of the mean market random effects on catch rates added (in log space) to the mean country-wide landings and then fit to market-level observed catches in each month. Note that Mobula thurstoni was not modelled with market-level random effects. (d) indicates months with survey effort (black squares) at each market.

Click here for additional data file.

We are grateful to the Department of Wildlife Conservation and the Department of Fisheries and Aquatic Resources for their advice and support. This work would not have been possible without dedicated researchers spending long hours at landing sites, and for this we thank Gobiraj Ramajeyam, Buddhi Maheshika, Sahan Thilakaratna, Lashanthini Rajendram, Rosalind Bown, Thejani Balawardana, Rifdha Rizwan, Tharushi Malshani, Gayathra Bandara, Thimali Dharmakeerthi, Stuart Monteith, and Katharina Gensch. We thank Akshay Tanna for database management, development of maps, and assistance provided for preliminary analyses. We also extend a special thanks to Dr Guy Stevens for helping initiate and support this project.

Additional Information and Declarations

Competing Interests

Author Contributions

Field Study Permissions

Data Availability

The authors declare there are no competing interests. Daniel Fernando and Joshua D. Stewart are uncompensated Associate Directors of The Manta Trust. Daniel Fernando is an uncompensated co-founder and director of Blue Resources Trust.

Daniel Fernando conceived and designed the experiments, performed the experiments, analyzed the data, prepared figures and/or tables, authored or reviewed drafts of the paper, obtained funding for the project, and approved the final draft.

Joshua D. Stewart analyzed the data, prepared figures and/or tables, authored or reviewed drafts of the paper, and approved the final draft.

The following information was supplied relating to field study approvals (i.e., approving body and any reference numbers):

All data was collected from public fishery landing sites. All specimens observed/recorded were dead fisheries specimens (from target or bycatch fisheries). Neither the fishery landing sites nor the species observed/recorded in this study are protected by any national laws at this point in time.

The following information was supplied regarding data availability:

Raw data are available in the Supplemental Files.

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
