# Peer review of "High bycatch rates of manta and devil rays in the “small-scale” artisanal fisheries of Sri Lanka"

_PeerJ, doi:10.7717/peerj.11994_

## Round 0.1 · original submission · Minor Revisions

Two reviewers provided helpful comments and critiques on this manuscript, and I agree with both reviewers that this is a valuable contribution to revealing potentially alarming trends in mobulid catch rates in a developing country. One reviewer recommends perhaps focusing the state-space models on the three species with the largest sample sizes to remove potential biases and re-examine the model output, as the extrapolated results seem high. I recommend, also as the reviewers suggest, to attempt to shorten the introduction and discussion to be more concise and improve the overall flow of the manuscript.

Reviewer 1 ·

Basic reporting

Please see general comments

Experimental design

Please see general comments

Validity of the findings

Please see general comments

Additional comments

This is an important study that provides an overall profile of mobulid bycatch in Sri Lankan artisanal fisheries over an 8-year period, as well as vital information on catch rates, annual landings and trends in body sizes for a number of mobulid species.

The manuscript is well-written and draws from an extensive long-term dataset to address knowledge gaps for threatened and understudied species, highlighting declines in catch rates and overfishing of vulnerable populations in the northern Indian Ocean. The findings provide a strong case for Sri Lanka and surrounding nations to improve management measures for these species in order to mitigate further declines.

I commend the authors on the amount of work and dedication that has gone into data collection for this study. The statistical methods used are sound and thorough and I cannot find fault with them. I do, however, find that the introduction and discussion sections are quite long and could be improved by reducing the amount of text and re-focusing to be more concise.

I therefore recommend minor revisions of the manuscript prior to acceptance. Please also see my specific comments below.

Line 76-77. Suggest removing “at present” to improve sentence flow. It is also unclear which fisheries are reliant on small-scale and artisanal fisheries, please re-word for clarity.

Line 106-107. “Studies on the oceanic manta ray (M. birostris) have also expanded
in recent years, resulting in important baseline data on movements and feeding ecology” please include the relevant citations here.

Line 165. Please include the definition for single-day and multi-day vessels here. I see this is explained in more detail further on in the manuscript but it would be useful here upon first mention.

Line 317. In which program/package did you implement the Bayesian Linear Regression? Please include this here.

Line 567. change ‘examination’ to ‘examinations’

Line 568. “. and” - full stop in the middle of the sentence. I also suggest reviewing the term “bycatch fishery”, perhaps change to “…provide an understanding of the scale of bycatch from this fishery/fisheries” or similar.

Line 612-613. With regard to the birostris catch numbers in Indonesia, over how long a time period were these numbers reported? Please add this.

Line 720. de Vos et al - is this the correct citation here? In fact, the citations in this paragraph seem to be mixed up (line 718-727), please review.

Line 727. Rohner et al 2017: there is no reference to match this is the reference list, is this supposed to be Stewart et al 2017?

Line 756-758. this information is a repeat from the introduction, consider re-wording to avoid repetition.

Reviewer 2 ·

Basic reporting

Meets basic reporting criteria.

Experimental design

Meets experimental design criteria.

Validity of the findings

Meets validity of findings criteria.

Additional comments

The Authors present a comprehensive paper on the mobulid fisheries in Sri Lankan artisanal fisheries. The findings are novel, well-presented and alarming. The methods employed and results obtained clearly indicate an unsustainable level of take of these species, albeit limited data for M. kuhlii and M. thurstoni. It also interesting that no M. alfredi were recorded given the wide-ranging nature of the fishers and their well documented occurrence in the general area (e.g. Maldives, Seychelles, Chagos). M. kuhlii's inferred coastal habitat use (like M. alfredi's) probably accounts for the small landings (or none) of these species. I wonder if for the state-space models it'd be worth focusing on pelagic-only fishery data, and on the three species with the larger sample sizes? This might remove some biases from the models, as some of the numbers seem a bit extreme. I am not 100% familiar with the methods employed, but the extrapolated numbers seem large.

The paper is robust and the fishery numbers are indicative of pervasive lack of enforcement and management for these species. This is difficult to implement, especially in cases where decision-makers and managers are often politically appointed and have political agendas to fulfil. I commend the authors for the work, and would encourage that this paper is presented to said managers and decision-makers in a simplified form that highlights the ongoing level of take and known unsustainable fishing trends. Perhaps the authors can add to the discussion, if there is, a more 'absolute' approach to indicate the level of take or the N value for the population at large. This would help get a better perspective on what the fishery numbers mean in terms of sustainability and conservation. Possibly SNPs or other genomic tools to indicate adult females at large within the subpopulation (n, fished), especially given their 1-2 pup strategy.

Some specific minor comments below:
Abstract
Line 20-21: consider not using ‘such as’ twice in the same sentence

Introduction
Line 44-45: use a ‘review’ reference here, or a few e.g. references
Line 67-68: if ‘significant’ give numbers. Can use considerable, and would still use a number (e.g. 50%).
Line 113: mobulid specific? State if so.
Line 115: ‘these’ species

Discussion
Line 568: punctuation typo here
Line 647: remove ‘with the exception of M. birostris’ as not being addressed here
Line 653: give species examples e.g. 10% size reduction for Carcharhinus falciformis etc
Line 724: >300 m

Figures
Table 1: consider this as a Supplementary Table as this is discussed/informed in the main text already (not specific numbers, but clear nonetheless)
Table 6: need maturity in the title “Size at maturity (DW) at which 50% of males…” or similar. Consider single decimal unit for consistency in the table.
Fig 1. Any reason no coverage in the southeast? Just curious.
Fig. 7. N values here would be good, perhaps in brackets after species name

---

## Round 0.2 · accepted · Accept

The authors have done an excellent job of addressing the comments of the reviewers and have streamlined the text significantly. I thank the authors and reviewers for their professional work during the review phase and accept this paper for publication. This will make a welcome contribution to understanding the impacts of small-scale fisheries in Sri Lanka and beyond.